# An Integrative Framework for Online Prognostic and Health Management Using Internet of Things and Convolutional Neural Network

**DOI:** 10.3390/s19102338

**Published:** 2019-05-21

**Authors:** Yuanju Qu, Xinguo Ming, Siqi Qiu, Maokuan Zheng, Zengtao Hou

**Affiliations:** 1SJTU Innovation Center of Producer Service Development, Shanghai Research Center for industrial Informatics, Shanghai Key Lab of Advanced Manufacturing Environment, Institute of Intelligent Manufacturing, School of Mechanical Engineering, Shanghai Jiao Tong University, Dongchuan Road 800, Minhang District, Shanghai 200240, China; xgming@sjtu.edu.cn (X.M.); siqiqiu@sjtu.edu.cn (S.Q.); zhengmaokuan@163.com (M.Z.); 2Shenzhen Institutes of Advanced Technology, Chinese Academy of Sciences, 1068 Xueyuan Avenue, Shenzhen University Town, Shenzhen 518055, China; zt.hou@siat.ac.cn

**Keywords:** prognostic and health management, integrative framework, internet of things, convolutional neural network, conditioned-based maintenance

## Abstract

With the development of the internet of things (IoTs), big data, smart sensing technology, and cloud technology, the industry has entered a new stage of revolution. Traditional manufacturing enterprises are transforming into service-oriented manufacturing based on prognostic and health management (PHM). However, there is a lack of a systematic and comprehensive framework of PHM to create more added value. In this paper, the authors proposed an integrative framework to systematically solve the problem from three levels: Strategic level of PHM to create added value, tactical level of PHM to make the implementation route, and operational level of PHM in a detailed application. At the strategic level, the authors provided the innovative business model to create added value through the big data. Moreover, to monitor the equipment status, the health index (HI) based on a condition-based maintenance (CBM) method was proposed. At the tactical level, the authors provided the implementation route in application integration, analysis service, and visual management to satisfy the different stakeholders’ functional requirements through a convolutional neural network (CNN). At the operational level, the authors constructed a self-sensing network based on anti-inference and self-organizing Zigbee to capture the real-time data from the equipment group. Finally, the authors verified the feasibility of the framework in a real case from China.

## 1. Introduction

Prognostic and health management (PHM) [1] is a reliable engineering approach that provides real-time health assessment and predicts its future state by using sensing technologies, machine learning, failure physics, etc. The main goal of PHM technologies is to provide the real-time health state of machines in order to improve the machine’s performance by taking proactive actions including diagnostics and prognostics [2,3]. The classification of prognostic models includes physical models, knowledge-based models, data driven models, and hybrid models [4]. The fault detection and failure progression was solved by the Kalman filter state-space predictor, and fuzzy logic classifiers method in an actuator case [5]. Additionally, PHM is usually studied in a laboratory without considering the influence of aging, the effect of people and a working environment, and the subject is usually a single component like gear, bearing and so on, which does not involve multi-sensor information fusion. In study [6], the authors used a 1D convolutional neural network (CNN) in a structural damage detection system. The prognostic algorithms were an effective method to solve fault prognosis in CBM systems for improving prediction accuracy and precision [7]. Meanwhile, Internet of Things (IoTs) is used in tracking, environment monitoring, sensing, and data collection in PHM [8], which is continually being adopted by the industry [9]. The PHM research was considered only from the application layer not involving the management and value layer. 

Therefore, the authors proposed an integrative framework including three levels: Strategic level, tactical level, and operational level to study the online PHM of heavy equipment by using IoTs and a two-layer CNN. The main aim was to provide more added value by effectively and efficiently managing the heavy equipment and using the data from massive sensors. In fact, the usage of massive amounts of sensors and IoTs appears only in recent years, and there are no sufficient data especially for heavy equipment whose lifespan is more than twenty years to train weights of the value from different sensors. The authors designed a self-sensing network based on Zigbee [10], which decreased energy consumption by automatically adjusting the transmitting speed of data according to the distance to the failure threshold. Additionally, this paper only focused on the forward transmission of CNN and the weights of sensors were determined by experts. 

The rest of this paper is organized as follows: Related research is reviewed in Section 2. The research methodology is illustrated in Section 3. A case study of PHM was carried out in Section 4. The results of implementing the integrative framework are analyzed in Section 5. Conclusions are presented in Section 6.

## 2. Literature Review

Prognostics and health management are widely used in the product, manufacturing environments, mainly including the monitoring, diagnostics, and prognostics from components, system, network, and related methods. Therefore, this section investigates different aspects of PHM through the previous research efforts. 

### 2.1. Component Layer of Prognostics and Health Management

A lot of researchers paid attention to components problems in PHM, such as the feature selection and management of rolling element bearings [11]. Byington et al. [12] developed a dynamic model of flight actuator to detect faults and predict failure for flight control actuators. Kacprzynski et al. [13] used statistical models to predict degradation rates of turbine compressor. Mba et al. [14] introduced a classification system of health state for a gearbox by integrating stochastic resonance method and hidden Markov modeling (HMM) method. Wang et al. [15] developed a stochastic degradation model to study the capacity degradation of batteries. Li et al. [16] used an ensemble learning method to predict the health degradation of aircraft engines. The studies usually focused on one type of part and there was no application of integrating a lot of sensors.

### 2.2. System Layer of Prognostics and Health Management

Some researchers studied the PHM of machines from a system aspect and promoted the synthetic application of sensors. Fitouhi and Nourelfath [17] solved a single machine’s integrating problem to provide preventive maintenance. Li et al. [18] developed an ensemble degradation model for engineering systems with multiple sensors using the health index synthesis (HIS) approach. Moghaddass and Zuo [19] proposed an integrated framework for a gradual degrading device using multistate stochastic process. Sensor systems [20] were used in the PHM to monitor operational, environmental, performance-related characteristics. Therefore, the number of sensors was small and the studies only referred to the operational level. Researchers have been making inroads into using emerging technologies to improve current practice, but it is not enough. 

### 2.3. Network Layer of Prognostics and Health Management

The network layer of PHM involves equipment to equipment communication, environmental sensing, online monitoring of key parts, and so on. Internet of Things and wireless sensor networks (WSNs) are widely used to get data in the industry [21]. Data aggregation is the strategy of wireless sensor networks [22], which is used in PHM for the communication of information between the equipment. Internet of Things is a network system that connects equipment with sensors, hardware network, and cloud servers [10,23]. Yang et al. [24] proposed a cloud-based prognostics system which was able to provide a low-cost solution for big data collected from a factory. Xia et al. [25] developed a condition monitoring system based on IoTs to resolve potential weaknesses. Korkua et al. [26] proposed a PHM system based on ZigBee to study rotor vibration under different conditions. Li et al. developed a real-time monitoring system for transport machines by using Radio Frequency Identification (RFID) and Global Positioning System (GPS) [27]. These methods only provided partial function applications in PHM and there is still a lack of multi-layer integrating research about PHM. 

### 2.4. Related Method of Prognostics and Health Management 

In past research, Bayesian networks [28], time domain analysis [29], gaussian mixture model [30], logistic regression [31], neural network [32], Kalman filter [33], and other algorithms were used in the PHM. Convolutional neural network is excellent in feature extraction of the data and is usually used as artificial intelligent algorithms through back propagation in the prediction of residual useful life (RUL) or fault recognition of parts, which needs a lot of historical data to train weights. Jing et al. [34] developed an innovative CNN for mechanical diagnosis to learn features directly from vibration signals. Jia et al. [35] proposed a local CNN to directly learn the health conditions of machines. Guo et al. [36] predicted bearings’ RUL by proposing a recurrent neural network based on a health indicator. Shao et al. [37] developed a deep CNN for rotating machines to provide accurate diagnosis of a certain part by fusing the monitoring data. Jia et al. [38] proposed an intelligent method based on CNN to predict the health condition of gearboxes and bearings. Chen et al. [39] employed three deep CNN models to identify the fault condition of rolling bearings based on the health condition. With the increasing development of IoTs and CNN in the field of processing massive data and things, more research efforts are needed to adapt to the development of the times [40,41]. Although the researchers promoted the development of PHM by using CNN, the results were usually not good because of the lack of data. To better use the CNN under this condition, the weights of the extracted features had to be determined reasonably by experts at the beginning.

### 2.5. Motivation and Objectives 

Therefore, based on the analysis of the literature review, an innovative framework for PHM that integrates CNN technologies and IoTs into current health management practices was proposed to systematically solve management questions of equipment group with little historical data. This paper aims to provide the systematic and comprehensive framework of PHM for the traditional manufacturers to provide continuous service to their customers, by integrating a two-layer CNN and self-sensing network to deal with the complex state of equipment group. This paper reports the first stage of the development, implementation, and evaluation of the framework which demonstrates how a two-layer CNN and Zigbee network can support the integrative framework for PHM innovation from strategy, tactic and operation levels, although not all of the features in the framework were developed in the current research.

## 3. Methodology 

Following the review of literature, a three-level framework based on CNN and IoTs technology was proposed (Figure 1), including the strategic level of PHM, tactical level of PHM, and operational level of PHM. The operational level of PHM which contained products, sensors, network, and database was in charge of the effective and efficient utilization of hardware by resisting interference and reducing energy consumption. The tactical level of PHM which integrated the advanced software and hardware to effectively and efficiently process data, mainly focused on the functional setting, including the visual management, analysis service, and application integrating function. The strategic level of PHM was in charge of defining goals and value types of different stakeholders, including value-added services and business system optimization. 

### 3.1. Strategic Level Innovation

The strategic level defines the companies’ objective of health management and creates a value-added method. Business system optimization and a value-added service helps companies reduce accidents and improve customer satisfaction by predicting service, continuously improving equipment performance in the design phase and extending equipment life. 

At this level, HI supported by the tactic level was proposed to reflect the health state. A condition-based maintenance (CBM) method was applied to improve equipment’s health state through real-time observation of HI. While HI decreases to a conditioned level, the maintenance will be carried out. The worst part of the equipment and the worst equipment of the batch which were important for the design optimization are also indicated by the framework at this level. A linear function (Equation (1)) was used as the display function by which the range of the HI was magnified from 1 to 100 and the change direction of the HI became positive to the equipment’s health state.
(1)$$f\left(x\right)=100*\left(1-\frac{x}{T}\right)$$
where, *x* represents the output value from max pooling layer or convolutional layer and *T* is the threshold value of *x*. 

### 3.2. Tactical Level Innovation

At this level, the authors provided application integrating, analysis service, and visual management module for the related stakeholder through CNN with two convolutional layers and two max pooling layers. The CBM-CNN was built to effectively extract the features for the purpose of data analysis, fault correlation analysis, timely detection of potential problems. The expression of the general activation function of CNN is shown in Equation (2).(2)y=f(W×X+B)
where, *W* represents the weight vector of sensors from equipment; *X* represents the vector of input value of real-time data from the equipment; *B* represents the bias of each sensor; *f* represents the activity function; *y* represents the health index value of each equipment. 

In this paper, the authors use a two-layer CNN to process the data from sensors (Figure 2). The input value *X* is convoluted through the equipment layer and the output layer and then sent to the display function with the max pooling value from *X* and equipment layer. The *f* in the convolution layer (CL) is a rectified linear unit (ReLU) function (Equation (3)).
(3)$$f\left(x\right)=\{\begin{array}{ll} 0, & x<0\\ x, & x\geq 0 \end{array}$$

The schematic diagram of the CNN is shown in Figure 3. Every row represents an equipment and every column represents the same sensor installed on different equipment. The relevant symbols represent the following meanings:

*x_mn_* is the value from the *n*th sensor installed on the mth equipment;

M*_m_* represents the synthetic health status of the mth equipment obtained from the first convolution of *X* and $W_{1}$;

$x_{mP}$ is the max pooling value of the mth row, which indicates the worst part of the mth equipment; 

$x_{Pn}$ is the max pooling value the *n*th column, which indicates the worst equipment for the same part;

$O_{x}$ represents the worst part of all equipment, which is the max pooling value of $x_{mP}$ or $x_{Pn}$; 

$O_{M}$ represents the worst health status of equipment, which is the max pooling value of M*_m_*; 

$O_{c}$ represents the synthetic health status of all equipment, which is the convolutional value of M and $W_{2}$;

Through the display function, $O_{x}$, $O_{M}$, and $O_{c}$ will be finally displayed in the control center.

### 3.3. Operational Level Innovation

Technologies of IoTs and smart connection satisfies the requirements of the sensing layer and network layer of PHM. To better resist interference and reduce energy consumption, a self-sensing network based on ZigBee was constructed and the topological structure is illustrated in Figure 4. 

In the ZigBee network, there are three kinds of nodes: Coordinator, router and end device. Every network has a coordinator which is the command center. The end devices are installed together with sensors, which are like the dendrites of neurons. The routers are just like the nucleus of neurons which collect data from end devices and communicate with each other. When a certain router breaks down, the corresponding end devices are able to communicate with nearby routers automatically. A self-sensing network is constructed to better balance the data requirements and energy consumption. The transmitting speed varies according to the value *x* of sensors and frequency *f* is shown in Equation (4).
(4)$$f=\max(\mathrm{int}(100*\frac{x}{T})/100,0.01)*f_{0}$$
where f is the transmitting frequency of the node; $f_{0}$ is the initial transmitting frequency of the system; T is the failure threshold of the corresponding component; *x* is the value of the sensor; ‘int’ means to retain the integer component of the value.

The transmitting speed of this node depends on the real-time value from sensors. When the value is nearer to the failure threshold, the transmitting speed of this node becomes faster so as not to miss important information. The transmitting frequency of the node varies from $0.01*f_{0}$ to $f_{0}$.

## 4. Case Study

To validate the feasibility of the proposed approach, a prototype was developed and tested by crane health management (CHM) practitioners. The current version named crane health management systems (CHMS) focuses on cluster health management and only has the ability to provide short-term forecast. The application was built in H company who is a big supplier of port equipment. With the saturation of the market, they wanted to change their role by CHMS innovation so as to create sustainable profits. The framework of CHMS is illustrated in Figure 5.

At the operational level, the important parts such as suspension bridge structure, motor and reducer were respectively monitored by corresponding sensors including stress sensor, vibration sensor, acceleration sensor, and so on. In the port, there were 10 cranes and every crane had 67 sensors installed (in Table 1). The designing and choosing principle of the placement of each sensor depended on the historical experience of the fault point and structure characteristic of the equipment. Displacement sensors were installed nearby the holes of the steel structure. Fiber optic acceleration sensors were installed on motors and gear boxes to measure vibration. Full bridge strain gauges were installed on the booms and their connections to measure strain and stress. Fiber Bragg grating temperature sensors were installed in motors and gear boxes to monitor temperature. 

ZigBee wireless network is applied as shown in Figure 6. Each end device was paired with one sensor and all equipment worked with one router. There were still several routers between the cranes and the local servers to amplify the signal and improve the robustness of the network. The data was saved in the local servers and copied to the cloud. The database in the cloud was based on the Hadoop, which was good for dealing with large data and assuring the integrity.

At the tactical level, the CNN with two convolutional layers and two max pooling layers were used to analyze the input data including producing HI, finding the worst part and finding the worst equipment. According to the experts, the weights and bias value of CHMS are listed in Table 2.

There was seven new equipment and three old equipment in the port. The input values at a certain time *t* to the different routers are listed in the Table 3.

At the strategic level, business optimization system and value-added service were carried out including performance guarantee, cluster optimization, equipment optimization, and so on (Figure 5). Using CHMS, the managers were able to easily make decisions ahead of time including component replacement, personnel transfer, manufacturing plan, and so on under the CBM principle which was that when the HI decreased by 5, the maintenance would be carried out.

## 5. Results and Discussion

Replacing the variables by values in Table 2 and Table 3, we obtained the values of different layers in CHMS as shown in Table 4.

$M_{m}$ were the machines’ HI through displaying function, seven of which were 95.9 and three of which were 60.0. $O_{x}$ and $O_{M}$ which represented the HI of the worst part and the worst machine were 58.8 and 60.0 respectively, and $O_{C}$ which represented the HI of the whole ten machines was 85.2. Because the normal HI of $M_{8}\sim M_{10}$ was 65, when it declined to 60.0, the principle of CBM was triggered and a maintenance warning was sent to the monitor and mobile terminal ahead of time. The partial interfaces of CHMS are shown in Figure 7, including real-time monitoring interface of port cranes, real-time monitoring interface of steel structures, real-time monitoring interface of a single port crane and real-time monitoring interface of lifting mechanisms. The real-time monitoring interface of port cranes contains Key Performance Indicator (KPIs) of CHMS and overall parameter information of port cranes. The real-time monitoring interface of steel structures displays the value of steel structures in different cranes including stress parameter, displacement parameter and strain parameter. The real-time monitoring interface of a single port crane contains all parameter information that can demonstrate the HI of a port crane. The real-time monitoring interface of lifting mechanisms shows the value of lifting mechanisms in different cranes including vibration parameter, stress parameter, temperature parameter and strain parameter.

After application of this framework in CHSM, the three old equipment were dumped and finally the residual useful life of the equipment was prolonged from 5 years judged by experiences to 6 years. Figure 8 shows the variation of health degradation with the application of this framework. Point A is the turning point of health degradation in CHMS. The past curve is based on history data and the expected curve is the prognostic curve based on the traditional method. In the new current curve, CHMS slowed down the health degradation and thus caused the extension of the RUL from $t_{1}$ to $t_{2}$, which was about 1 year.

Apart from the extension of the RUL, there were other improvements for the stakeholders’ value. Table 5 presents the comparison before and after implementing the framework.

With the application of CHMS, more added value is achieved for the manufacturer and the end user. Although the prognostic ability of the system is weak now, with the accumulation of data in such a framework, the training ability of CNN will become true and thus the system can be better in the future. 

## 6. Conclusions

Prognostic and health management is an important method for manufacturers in order to monitor failure precursor, improve product performance, and create added value. Examining the existing literature related to PHM, this paper proposed an integrative framework of PHM based on IoTs and CNN through practical investigation. The framework provided the systematic guidance of PHM for manufacturers from three levels: Strategic level, tactical level and operational level, which would help more companies build a win-win relationship and create more added value, such as the transition from selling products to selling services, continuously improving product performance, and continuously improving customers’ satisfaction. At the same time, this paper also plays an exemplary role in showcasing the usage of CNN and IoTs in the fusion of massive sensors. The contributions of this paper were concluded as follows:This paper provided an integrative framework for PHM to give a new business model for traditional manufacturers in expanding innovative business model and achieving added value from three levels.At the strategic level, the HI was proposed and used with CBM to provide value-added service and business chance referring to optimize product performance and reduce the operation and maintenance cost.At the tactical level, the authors developed a two-layer CNN with reasonable weights achieving more added value by effectively and efficiently managing the heavy equipment and using the data from massive sensors.At the operational level, this paper proposed the self-sensing network based on Zigbee to realize the monitoring of the real-time data from the equipment group.The case study of CHMS was proposed in this paper to check the feasibility of the framework from the value added and prediction of residual useful life of heavy equipment.

Although the proposed integrative framework demonstrates potentials in PHM, there is still more research that needs to be done in the future. To cope with real world practices, more data needs to be collected under this framework to optimize the weights and reveal the relationship between the HI and the RUL through deep learning. Future research will focus on this area.

## Figures and Tables

**Figure 1 sensors-19-02338-f001:**
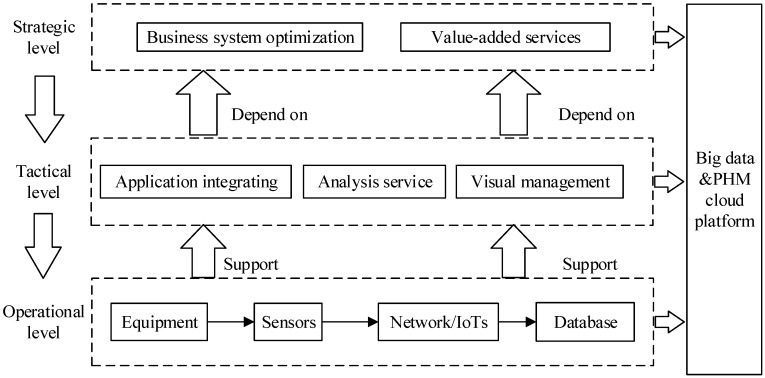
Integrative Prognostics and Health Management framework base on CNN and IoTs.

**Figure 2 sensors-19-02338-f002:**
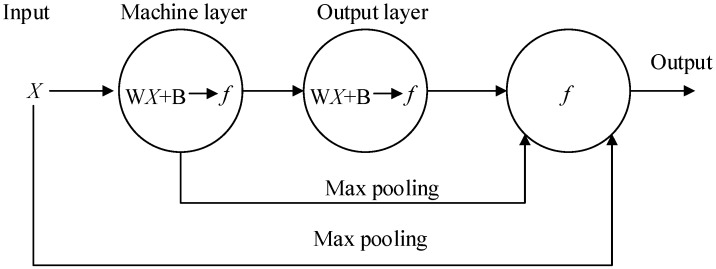
Architecture of the CNN.

**Figure 3 sensors-19-02338-f003:**
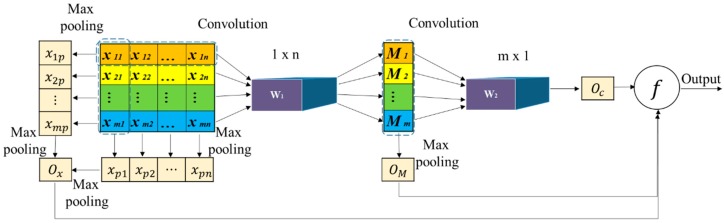
Schematic diagram of the CNN.

**Figure 4 sensors-19-02338-f004:**
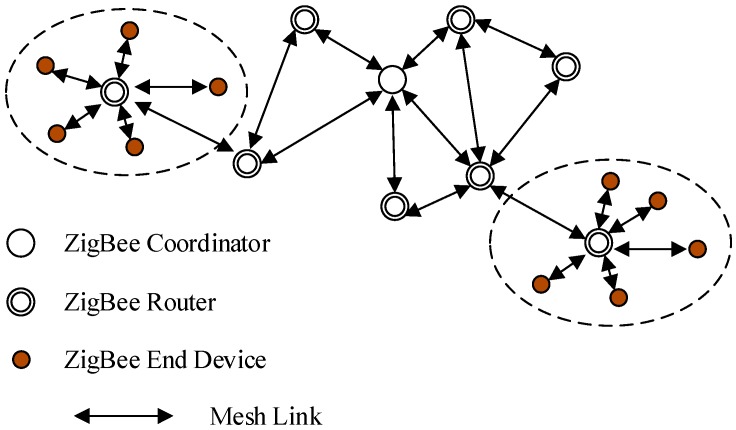
ZigBee topological structure.

**Figure 5 sensors-19-02338-f005:**
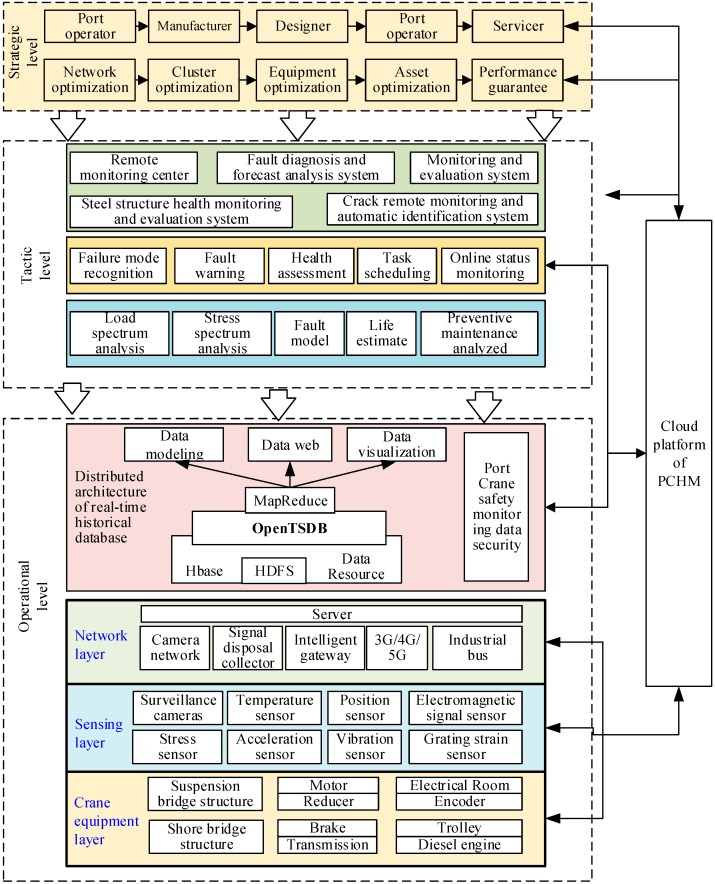
Application of the innovative CHMS framework.

**Figure 6 sensors-19-02338-f006:**
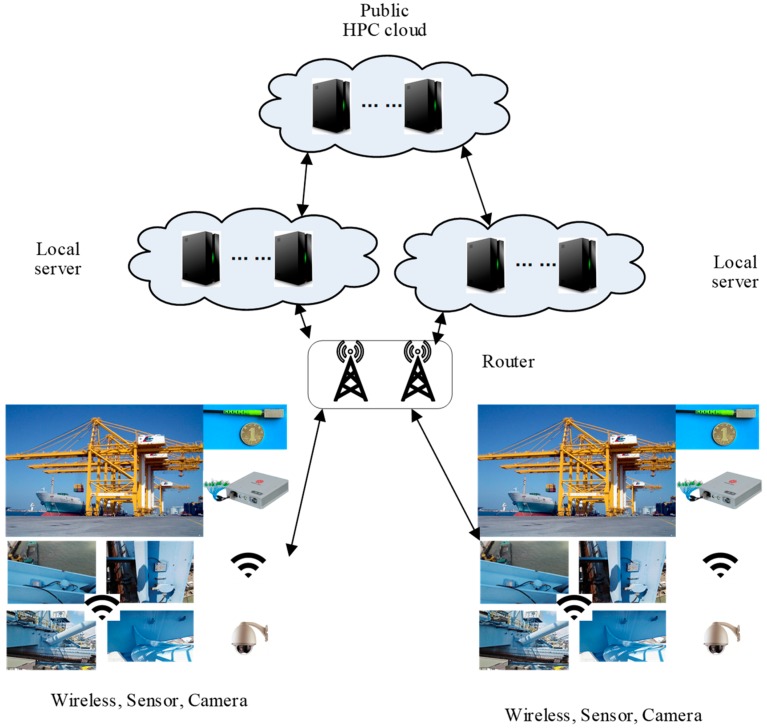
ZigBee wireless network.

**Figure 7 sensors-19-02338-f007:**
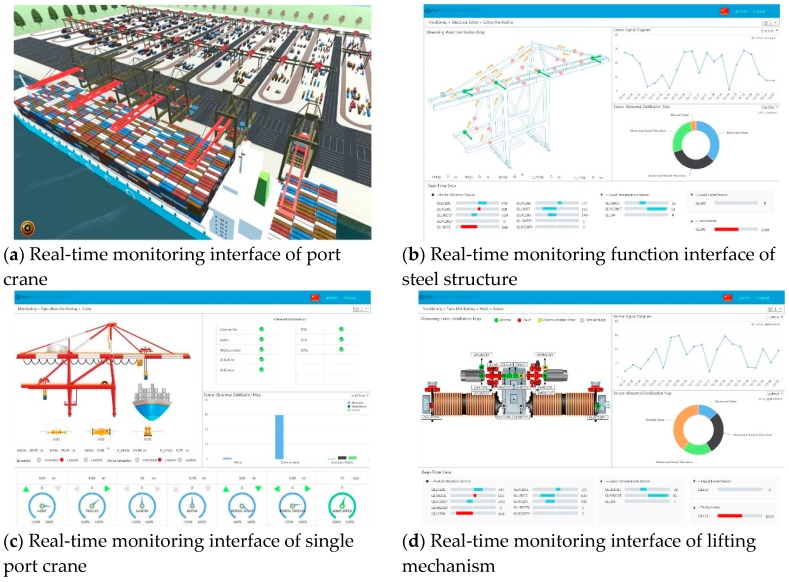
Application interfaces of CHMS.

**Figure 8 sensors-19-02338-f008:**
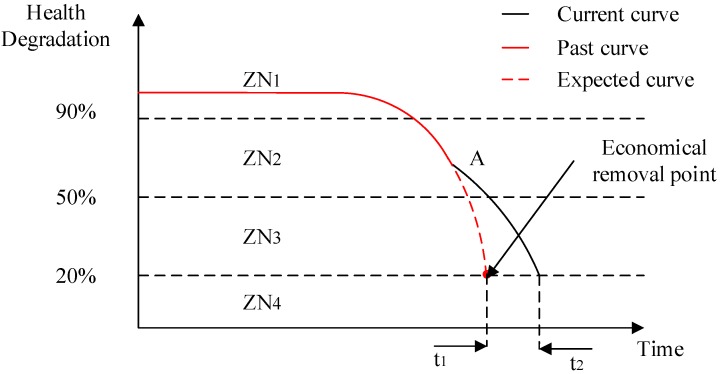
The variation of health degradation. Notes: ZN1 is the safe area; ZN2 is the wearing area; ZN3 is the high-risk area; ZN4 is the scrap area.

**Table 1 sensors-19-02338-t001:** Sensors installed in each equipment.

Symbol	Description
$X_{1}\sim X_{32}$	Displacement sensors need to be arranged at steel structure whose threshold value is 20 mm.
$X_{33}\sim X_{38}$	Stress sensors need to be arranged at different places that are $X_{33}\sim X_{38}$, whose threshold value is 100 MPa
$X_{39}\sim X_{40}$	Strain sensors need to be arranged at different places whose threshold value is 3 mm.
$X_{41}\sim X_{50}$	Axial vibration sensors need to be arranged at different places whose threshold value is 2 mm.
$X_{51}\sim X_{54}$	Radial vibration sensors for main motors need to be arranged at different places whose threshold value is 15 mm.
$X_{55}\sim X_{60}$	Radial vibration sensors for small motors need to be arranged at different places whose threshold value is 10 mm.
$X_{61}\sim X_{67}$	Temperature sensors need to be arranged at gear boxes whose threshold value is 95 °C.

**Table 2 sensors-19-02338-t002:** Weights and bias of the CNN for CHMS.

Weights of Sensors	$W_{1}$	$w_{11}\sim w_{132}$	$w_{133}\sim w_{138}$	$w_{139}\sim w_{140}$	$w_{141}\sim w_{150}$	$w_{151}\sim w_{154}$	$w_{155}\sim w_{160}$	$\omega_{1161}\sim \omega_{1167}$
		1/20	1/100	1/3	1/2	1/15	1/10	1/30
Bias of sensors	$B_{1}$	$b_{11}\sim b_{160}$				$b_{161}\sim b_{167}$		
		0				−2.2		
Weights of crane	$W_{2}$	$w_{21}\sim w_{210}$						
		1/67						
Bias of crane	$B_{2}$	$b_{21}\sim b_{210}$						
		0						

**Table 3 sensors-19-02338-t003:** Matrix of the input values from sensors to different equipment in CHMS.

*X*		Ten Equipment in CHMS	
		$X_{M1}\sim X_{M7}$	$X_{M8}\sim X_{M10}$
Value of 67 sensors	$x_{1}\sim x_{32}$	1.0	8.0
	$x_{33}\sim x_{38}$	1.7	41.1
	$x_{39}\sim x_{40}$	0.2	1.2
	$x_{41}\sim x_{50}$	0.1	0.7
	$x_{51}\sim x_{54}$	0.7	6.1
	$x_{55}\sim x_{60}$	0.4	3.9
	$x_{61}\sim x_{67}$	48.0	77.0

**Table 4 sensors-19-02338-t004:** HI of the three layers in CHMS.

Symbol	$M_{1}\sim M_{7}$	$M_{8}\sim M_{10}$
$M_{m}$	95.9	60.0
$O_{x}$	58.8	
$O_{M}$	60.0	
$O_{C}$	85.2	

**Table 5 sensors-19-02338-t005:** Comparison before and after implementing the framework in CHMS.

Stakeholder	Item	Before	After	Main Contribution
Company H	Business model	Sell equipment to the port and time-based maintenance	Condition based maintenance and provide health management service, fault prognostic and diagnostic service, breakdowns avoiding service for charges	Business model transition from manufacturer to service provider; Work enthusiasm is incented by reducing personnel and increasing salary; Supply better product based on the history data. Reducing the waiting time.
	Personnel	2	1	
	Salary	Every worker earns RMB 80,000 per year	Every worker gets as much as RMB 90,000	
	Added value	0	Nearly RMB 100,000	
End user	Personnel	3	2	The labor and downtime costs decrease a lot and the productivity rises greatly
	Breakdowns	More than 10 times per year	0	
	Fault rate	About 5% every year	Less than 1%	
	Residual useful life	According to experience, the old equipment’s RUL is about 5 years	By the framework, the RUL of the old equipment is prolonged to about 6 years	

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
