# Peer review of "An Integrative Framework for Online Prognostic and Health Management Using Internet of Things and Convolutional Neural Network"

_sensors, 2019, doi:10.3390/s19102338_

Round 1
Reviewer 1 Report
The paper has been improved. Please read again the whole paper and correct some english errorr.
Author Response
Comments#1: The paper has been improved. Please read again the whole paper and correct some english errorr. # Answer and action: Thank you for your valuable comments and recognition of this paper. We have checked the English language and corrected related sentences again.
Reviewer 2 Report
The manuscript improved slightly, but it still contains many problems in the presentation making the readability tedious. In the following I will give some examples:
Line 19: Do you mean "systematically solve"?
Lines 22-23: A bit repetitive use of "based on, based on". Would be better to split the sentence into two maybe.
Line 24: Do you mean "integration"?
Line 25: Remove the "open-loop", does not explain nothing here.
Line 28: Instead of "Ultimately" maybe better use "Finally".
Line 40: "in an actuator case".
Line 40: "in a laboratory".
Line 41: "fault" -> "effect".
Line 42: "part" -> "component".
Line 43: "1D convolutional neural network (CNN) [6] is used in structural damage detection system.". Did you mean to say: "In study [6], the authors used 1D convolutional neural network (CNN) in structural damage detection system.".
Line 44: "are an effective method".
Line 52: What is a "massive sensor"? It is unclear do you actually mean physically large sensors, or do you mean "massive amounts of sensors".
Line 54: "weights of sensors" What does this mean?
Line 55: Define "Zigbee" or add reference.
Line 56: "the threshold value" what is this value?
Line 68: Do you mean "Component layer of PHM"?
Section 2.1: Reference style changed? Compare to introduction.
Line 155: What is the "normalization of x"? Define, be more clear.
Equation 2: This definition is too general, it contains very little information. You could have rather written that equation 2 represents the general activation function of CNN.
Line 174: "gotten", check this but I think "got" is better here.
Figure 3: Make the figure more clear, use vector graphics if possible or use higher pixel density image. Make the red color more brighter, not reader friendly.
Line 227: So "~" means "to" according to the response letter. Add this to the manuscript. The symbol "~" has many interpretations depending on the context. It can mean e.g. "distributed as", "approximately". This is meaning is not now clearly stated in the text.
As I stated before, the research design of the manuscript is valid and interesting, but due to the multiple shortcomings in the presentation a major revision must be implemented before reconsideration. I highly recommend the authors consult the help of a professional English language expert before next re-submission.
Author Response
Dear Reviewer,
Thank you very much for your letter and for reviewers’ comments concerning our manuscript entitled “An integrative Framework for online Prognostic and Health Management using Internet of Things and Convolutional Neural Network”. These comments are significant to the improvements of our article. The refined contents are marked by yellow color in that revised manuscript. We hope that our modifications and refinements can meet your approval. The specific answers to reviewers’ comments are stated as follows, in which ‘Black font with underline’ represent ‘Reviewer’ Comments’, while ‘Red font’ represent ‘Answers’.
Thanks again and Best Regards.
Sincerely yours,
Yuanju Qu
Reviewer 2
Comments #1: The manuscript improved slightly, but it still contains many problems in the presentation making the readability tedious. In the following I will give some examples:
# Answer and action: Thank you for your valuable comments and recognition of this paper. We have checked the paper to avoid similar problems.
Comments #2: Line 19: Do you mean "systematically solve"?
# Answer and action: Thanks for your suggestion, we have change “systematic solve” to “systematically solve”
Comments #3: Lines 22-23: A bit repetitive use of "based on, based on". Would be better to split the sentence into two maybe.
# Answer and action: Thanks for your suggestion, we have split the sentence into two: “At the strategic level, the authors provide the innovative business model to create added value through the big data. And to monitor equipment status the health index (HI) based on condition-based maintenance (CBM) method is proposed”.
Comments #4: Line 24: Do you mean "integration"?
# Answer and action: Thanks for your suggestion, we have change “integrating” to “integration”.
Comments #5: Line 25: Remove the "open-loop", does not explain nothing here.
# Answer and action: Thanks for your suggestion, we have deleted “an open-loop” it.
Comments #6: Line 28: Instead of "Ultimately" maybe better use "Finally".
# Answer and action: Thanks for your suggestion, we have change “Ultimately” to “Finally”.
Comments #7: Line 40: "in an actuator case".
# Answer and action: Thanks for your suggestion, we have change “in a actuator case” to “in an actuator case”.
Comments #8: Line 40: "in a laboratory".
# Answer and action: Thanks for your suggestion, we have change “in laboratory” to “in a laboratory”.
Comments #9: Line 41: "fault" -> "effect".
# Answer and action: Thanks for your suggestion, we have change “fault” to “effect”.
Comments #10: Line 42: "part" -> "component".
# Answer and action: Thanks for your suggestion, we have change “part” to “component”.
Comments #11: Line 43: "1D convolutional neural network (CNN) [6] is used in structural damage detection system.". Did you mean to say: "In study [6], the authors used 1D convolutional neural network (CNN) in structural damage detection system.".
# Answer and action: Thanks for your suggestion, we have change “1D convolutional neural network (CNN) [6] is used in structural damage detection system.” to “In study [6], the authors used 1D convolutional neural network (CNN) in structural damage detection system”.
Comments #12: Line 44: "are an effective method".
# Answer and action: Thanks for your suggestion, we have change “are effectively method” to “are an effective method”.
Comments #13: Line 52: What is a "massive sensor"? It is unclear do you actually mean physically large sensors, or do you mean "massive amounts of sensors".
# Answer and action: Thanks for your suggestion, we have change “massive sensor” to “massive amounts of sensors”.
Comments #14: Line 54: "weights of sensors" What does this mean?
# Answer and action: Thanks for your constructive suggestions. We mean that every value of the sensor has a weight in the convolutional calculation of CNN just as shown in equation 2 and figure 2. To better express our meaning, we change it to “weights of the value from different sensors”.
Comments #15: Line 55: Define "Zigbee" or add reference.
# Answer and action: Thanks for your constructive suggestions. According to your suggestions, we have added reference in the introduction such as “The authors design a self-sensing network based on Zigbee [10], which decreases energy consumption by automatically adjusting the transmitting speed of data according to the distance to the failure threshold.” Highlight in yellow colors.
Comments #16: Line 56: "the threshold value" what is this value?
# Answer and action: Thanks for your suggestion, we have changed it to “the failure threshold”.
Comments #17: Line 68: Do you mean "Component layer of PHM"?
# Answer and action: Thanks for your suggestion, we have change “Components layer of PHM” to “Component layer of PHM”.
Comments #18: Section 2.1: Reference style changed? Compare to introduction.
# Answer and action: Thanks for your suggestion, we have changed the reference style into “MDPI” style according to the “MDPI reference list and citation style guide”.
Comments #19: Line 155: What is the "normalization of x"? Define, be more clear.
# Answer and action: Thanks for your suggestion, we have change it to “n(x)=x/T and T is the threshold value of x”. The modified part is presented in Line 133-134 on Page 7 in Section 3.1.
Comments #20: Equation 2: This definition is too general, it contains very little information. You could have rather written that equation 2 represents the general activation function of CNN.
# Answer and action: Thanks for your suggestion, we have change the sentence to “The expression of the general activation function of CNN is shown in equation (2)”. The modified part is presented in Line 139-144 on Page 8 in Section 3.2.
Comments #21: Line 174: "gotten", check this but I think "got" is better here.
# Answer and action: Thanks for your suggestion, we have change “gotten” to “got” in Line 156 on Page 8 in Section 3.2.
Comments #22: Figure 3: Make the figure more clear, use vector graphics if possible or use higher pixel density image. Make the red color more brighter, not reader friendly.
# Answer and action: Thanks for your suggestion, we have changed the Figure3 to clear one highlighted in yellows.
Comments #23: Line 227: So "~" means "to" according to the response letter. Add this to the manuscript. The symbol "~" has many interpretations depending on the context. It can mean e.g. "distributed as", "approximately". This is meaning is not now clearly stated in the text.
# Answer and action: Thanks for your constructive suggestions. According to your suggestions, we have added the "~" means "to" in the catalog of Acronyms in Page 2.
Comments #24: As I stated before, the research design of the manuscript is valid and interesting, but due to the multiple shortcomings in the presentation a major revision must be implemented before reconsideration. I highly recommend the authors consult the help of a professional English language expert before next re-submission.
# Answer and action: Thank you for your valuable comments and recognition of this paper. We have consulted the help of a professional English language expert and revised the English language again.

Round 2
Reviewer 2 Report
The paper has been improved accordingly and can be considered for publication.
Author Response
thank you very much for your comments. we have checked the language again, and now there is no problem.
This manuscript is a resubmission of an earlier submission. The following is a list of the peer review reports and author responses from that submission.
Round 1
Reviewer 1 Report
The work proposes a systemic study of a zigbee neural network improved by convolution neural network algorithm. About my opinion more elements should be discussed in order to provide originalities according with the state of the art [10] and [11]
A Major revision is required.
It is no clear the data process of Fig. 3: do sensors refer to different apparatus? How these sensors are correlated in the case of proposed study? Please explain better in the case of study how is applied the convolution. It is important to show better the case of study by designing the placement of each sensors and by indicating the possible correlation between values temperature, vibration etc.
It is important to fix a big data system in order to show some performance results about the computational cost also referring to a random values.
More information about the server backend system are required.
Please provide more information about neural network algorithm performance by hypotizing different training dataset models?
Reviewer 2 Report
The manuscript proposes a method for real-time prediction of the health condition of complex machines via internet of things and convolutional neural network (CNN) approach. The subject of the article is good and relevant for the journal, but it suffers from extensive problems in the English language, with many poorly written sentences and paragraphs. Also the methods, e.g. the CNN part is not well described. The mathematical symbols and acronyms are not clearly defined and listed, and while reading the manuscript I get lost in the large jungle of acronyms and unclearly defined symbols. To give some examples, I will next give detailed comments:
- Line 15: Did the authors mean "studies" here?
- Line 15: Do you mean "massive amounts of sensors"?
- Line 15: proposed -> propose
- Line 20: "consuming" -> "consumption"
- Line 30: Check the reference style. Not sure if this is the correct syntax to make citations, but I have never seen superscripts been used for this purpose before. If change is needed apply it to whole text.
- Line 34: "connect" -> "connects"
- Line 35: "industry" -> "the industry"
- Line 36: Check the English here. I'm not a native speaker but this seems to be not well written.
- Line 38-39: "better balance the data requirements and energy consuming.". First of all, this is identical to the one stated in abstract, intro shouldn't have identical parts as abstract in well written text. Secondly, change "consuming" to "consumption".
- Line 53-54: "are determined by experts but not black box.". Check the English here.
- Line 55-56: I can't understand this sentence. Check the English .
- Line 57: "proposed" -> "propose"
- Lines 59-61: Maybe list the questions in a e.g. numbered list. Change "component" -> "components"
- Lines 62-65: "was" -> "is", "were" -> "are"
- Section 2.4: The beginning is not written well by listing the studies in single and short sentences. I think better class writing should be applied here. A lot of tautology is also found with the use of "neural network".
- Line 107: "neural network" -> "a neural network"
- Section 3: Suffers from the same problem as section 2.4. Many sequential short sentences is not a good practice in general.
- Figure 2: Define "RFD" and "FFD". Readers familiar with these can always fast forward.
- Line 168: Maybe "lifespan" better than "lives"
- Equation 2 and lines 176-179: What is the definition of n? w_ij should be explained more, it's not enough to just state "w_ij represents the weight". So the function f receives as input numbers from the interval 0 to 1? Maybe clarify why this is relevant in the text. What does subscripts i and j stand for?
- Figure 3: Explain what the inputs x_abc stand for. What are meanings of all those subscripts? What is R_ij? What is S_k?
- Line 182: Not so good English, check sentence.
- Line 184: "if" -> "If"
- Equation 5: "1>1"?
- Line 206: Define "HI"
- Line 227: What does "~" mean?
The authors' approach is justified and relevant, but because of the major lacks in presentation, both language and technical description, I suggest the manuscript should be rejected in it's present form. A major revision is a minimum step to be taken next. First of all, the English language must be checked and corrected to make the text readable. Secondly, the presentation of the technical parts must be clarified with all relevant symbols defined when they are introduced.